# Advances in Integration, Wearable Applications, and Artificial Intelligence of Biomedical Microfluidics Systems

**DOI:** 10.3390/mi14050972

**Published:** 2023-04-29

**Authors:** Xingfeng Ma, Gang Guo, Xuanye Wu, Qiang Wu, Fangfang Liu, Hua Zhang, Nan Shi, Yimin Guan

**Affiliations:** 1School of Communication and Information Engineering, Shanghai University, Shanghai 200000, China; 2Department of Microelectronics, Shanghai University, Shanghai 200000, China; 3Shanghai Industrial μTechnology Research Institute, Shanghai 200000, China; 4Shanghai Aure Technology Limited Company, Shanghai 200000, China; 5Institute of Translational Medicine, Shanghai University, Shanghai 200000, China

**Keywords:** microfluidics, integration, miniaturization, artificial intelligence

## Abstract

Microfluidics attracts much attention due to its multiple advantages such as high throughput, rapid analysis, low sample volume, and high sensitivity. Microfluidics has profoundly influenced many fields including chemistry, biology, medicine, information technology, and other disciplines. However, some stumbling stones (miniaturization, integration, and intelligence) strain the development of industrialization and commercialization of microchips. The miniaturization of microfluidics means fewer samples and reagents, shorter times to results, and less footprint space consumption, enabling a high throughput and parallelism of sample analysis. Additionally, micro-size channels tend to produce laminar flow, which probably permits some creative applications that are not accessible to traditional fluid-processing platforms. The reasonable integration of biomedical/physical biosensors, semiconductor microelectronics, communications, and other cutting-edge technologies should greatly expand the applications of current microfluidic devices and help develop the next generation of lab-on-a-chip (LOC). At the same time, the evolution of artificial intelligence also gives another strong impetus to the rapid development of microfluidics. Biomedical applications based on microfluidics normally bring a large amount of complex data, so it is a big challenge for researchers and technicians to analyze those huge and complicated data accurately and quickly. To address this problem, machine learning is viewed as an indispensable and powerful tool in processing the data collected from micro-devices. In this review, we mainly focus on discussing the integration, miniaturization, portability, and intelligence of microfluidics technology.

## 1. Introduction

Microfluidics is a multidisciplinary technique that processes or manipulates $10^{-9}$~$10^{-18}$ liters of fluids with micrometer-size channels [1]. A microfluidic platform paves an effective way for the automatic and high-throughput analysis of biochemical samples [2]. According to the actuation mechanisms, microfluidic platforms can be classified into several groups, such as capillary, pressure-driven, centrifugal, electrokinetic, and acoustic [3]. Based on the fluid-propulsion force, microfluidics can be divided into continuous-flow and droplet microfluidics [4].

The liquids flow in the fabricated microchannels without breaking continuity in the continuous-flow microchips and they have evolved rapidly in the last decades [5]. However, some inherent issues (such as Taylor dispersion, solute–surface interactions, and cross-contamination) limit the further development of this type of microfluidics [6]. Droplet microfluidics is a type of technique that utilizes immiscible multiphase flow to generate and manipulate discrete droplets in the micro-size channels [7]. Microfluidic devices can produce discrete droplets at frequencies ranging from a few Hz to thousands of Hz. Each droplet is viewed as a microreactor encapsulating biochemical reactants, cells, nanoparticles, nucleic acids, or other components, and individually manipulated [8,9] For biological and chemical analysis, droplet reactors offer significant advantages of miniaturization and independence that represent the benefits of low consumption and low cross-contamination [10]. On the one hand, the miniaturization of the reactor makes single-cell or molecular analysis possible [11,12]. On the other hand, small droplets have a large surface area-to-volume ratio, which facilitates heat and mass transfer and accelerates the reaction rate [13]. The large number of uniform droplets generated by the microfluidic platform provides the possibility of parallel processing, paving the way for high-throughput analysis and enabling high-throughput screening under a variety of experimental conditions [14,15].

In the 1990s, many researchers explored the application of micro-electromechanical systems (MEMS) technology in the fields of biology, chemistry, and biomedicine, which lead to the generation of microfluidics. It is a versatile platform that integrates sample preparation, reaction, separation, and detection on a micron-scale chip [16]. In conclusion, building biochemical systems on the surface of a solid chip produced via MEMS technology enabled the fast and accurate processing and detection of proteins [17], cells [18], nucleic acids [19], metabolites [20], and other specific targets. Normally, the fabrication of microchips primarily relies on the microchip design, optical materials, and MEMS technologies. The purpose of the microchip determines the whole microchip design, such as the channel shape, channel size, position of valves, detection zone, micro-pumps, method of detection, chip size, etc.

In addition, the properties of fluids in microchannels cannot be ignored during the microchip design. The Reynolds number ($R_{e}$) reflects the ratio of inertial forces to viscous forces [21]. Within the microscale channels, the Reynolds number is usually less than 1 and the fluidic behavior is mainly influenced by the viscosity rather than inertia, which means the fluid flow is essentially laminar [22], and this specific feature is usually used in the generation of concentration gradients [23,24,25], macroscopically, polymer microrods, Janus droplets, etc. [26,27]. The Peclet number ($P_{e}$) represents the ratio between advection and diffusion. When the Peclet number is relatively large, the diffusion effect is weak, and the mixing becomes very difficult at this moment [28], which means the time for complete mixture should be very long even if the diffusion distance is short. This is critical for some reactions and measurements that have to be finished in milliseconds. This is the reason why some sinuous microchannels are required when designing microchips [29,30], which helps induce chaotic advection and accelerate fluid mixing. In addition, fabricating interconnected microchannel networks plays a key role in the construction of microchips where MEMS (photolithography) processes and optical materials are used to generate micron-size channels in silicon and other substrates [31].

The advent of LOCs miniaturizes benchtop laboratory equipment into portable tools for point-of-care (POC) diagnostics, while achieving orders-of-magnitude reductions in hardware costs and sample usage [32]. Although existing discrete devices have the ability to reduce the sizes of physical equipment, reasonable integration of microelectronics and integrated circuit techniques will further miniaturize microchips and effectively improve their performance [33]. When integrated circuits (ICs) are inserted into microfluidic platforms, a complementary metal-oxide semiconductor (CMOS) is able to distinguish and gather the weak changes of electrical signals (current, voltage, or electromagnetic wave) that are generated in some biochemical reactions and promotes the implementation of real-time quantitative analysis [34]. The integration of microfluidics and CMOS not only contributes to the portability of clinic equipment, but also helps perform rapid diagnostics for patients in non-laboratory and non-specialist conditions [35].

As per the description above, CMOS and MEMS techniques have contributed to the development of microfluidics [35]. In return, microfluidics as an emerging discipline also provides some creative ideas for the development of traditional semiconductor technology. The miniaturization of electronic devices and circuits causes the phenomenon of self-heating, which restricts in the performance and reliability of microelectronic electrical systems [36]. To solve this problem, microchannels have attracted the attention of engineers^,^ and manifold microchannel cooling systems were produced by putting microfluidic and electronic devices on the same semiconductor substrate, which greatly reduced the temperature of the semiconductor and the energy consumption for cooling electronic devices [37]. In summary, microfluidics and semiconductors naturally complement each other.

How to collect and process the huge data from droplet microchips is another tricky challenge for researchers [38]. Intelligent microfluidics is beginning to receive increasing attention from researchers as an emerging interdisciplinary research field that combines microfluidics with machine learning (ML), taking full advantage of the high throughput and controllability of microfluidics while introducing the powerful data processing capabilities of ML. In recent years, the possibility of combining machine learning and microfluidics has been verified in successfully solving biomedical and biotechnological issues [39]. By selecting appropriate training models and proper training, ML has proven to be a powerful tool for rapid and accurate feature extraction, classification, prediction, and optimization of the large amount of data generated in microfluidic systems [40]. Compared with traditionally manual analysis, intelligent microfluidics requires less human intervention, dynamically improving computer-aided prediction performance from large amounts of data.

Although microfluidics has been successfully applied in many fields until now there should be great potential to be exploited with the development of technology. Miniaturization, integration, and intelligence will probably provide a larger stage for microfluidic techniques. In this review, we first introduce some classical electrochemical biosensors and applications after integrating them with microfluidic devices. In the next part, wearable microfluidics is discussed, which includes multiple technologies such as sensors, flexible electronics, mobile communication, and other necessary elements for various purposes. Smart microfluidics with artificial intelligence technologies is investigated in the last section. This review gives some useful hints to researchers who are interested in microfluidic miniaturization, integration, and intelligence.

## 2. Integrated Microfluidic Systems

Integrated microfluidic system have the specific characteristics of integrating multiple analytical processes in a signal device [41]. The external physical techniques (such as optical, electrical, acoustic, and magnetic fields) that are introduced into microchips help make the integrated microfluidic system become a powerful platform for biomedical analysis [42]. Specifically, the combination of microfluidics and various electrochemical biosensors has also attracted researchers’ interests, which is viewed as an ideal way to implement the next generation of highly integrated and miniaturized LOC systems. Therefore, this chapter will briefly introduce the combination of external fields and microfluidic systems and finally focus on presenting the integrated electrochemical microfluidics.

### 2.1. Microfluidics Integrated with External Techniques

#### 2.1.1. Magnetic Fields and Microfluidics

The integration of microchips and magnetics has been demonstrated in many fields. Magnetic materials have been used as valves to control the status of fluids, and magnetic particles have been employed for mixing fluid streams. The most conventional application of magnetics in microfluidics is to separate target cells from suspensions [43,44].

Masumi Yamaha et al. presented a type of cell-sorting microfluidic system based on size and surface markers. This simple system sorted cells with high precision via combining the hydrodynamic filtration (HDF) scheme and magnetophoresis. As shown in Figure 1a, cells were first aligned onto the low sidewall in the main channel and then sorted into each separation lane (up to six) based on different sizes in the HDF scheme. Then the magnets were perpendicularly added to the separation lanes and sorted the similar-sized cells with different numbers of markers (up to four in each separation lane). In this microchip, the cells conjugated with more immunomagnetic beads would get closer to the magnet than those with fewer beads, thus achieving the second-round screening [45].

Karabacak et al. redesigned CTC-iChip to isolate circulating tumor cells (CTCs) from the whole blood with the assistance of magnetic force, and CD66 was used as a leukocyte marker in this system. As shown in Figure 1b, two independent chips with different modules were integrated to achieve screening CTCs. In the CTC-Chip-1, the white blood cells (WBCs) and tumor cells were separated from the whole blood based on the size with continuous deterministic lateral displacement (DLD). In the CTC-Chip-2, inertial focusing was used to position these cells in the microchannel precisely and the tumor cells were isolated through microfluidic magnetophoresis. This system achieved an average of 3.8 g-log depletion of WBCs with a sample processing rate of 8 mL/h, and the yield of rare cancer cells was 97 ± 2.7% [46].

#### 2.1.2. Acoustic Fields and Microfluidics

Microfluidics with acoustics is a powerful tool for manipulating particles and cells in biomedical applications. The devices have the properties of versatility, biocompatibility, precision, flexibility, compactness, and cost-effectiveness [47]. Generally, acoustic waves can be classified into two types: surface acoustic waves (SAWs) and bulk acoustic waves (BAWs) [48].

SAWs are often generated within piezoelectric materials, which can be exploited to screen particles or cells in a micro-system [49]. Lee et al. proposed putting four different modules in a microchip that utilized acoustic and electric fields to sort particles and cells. This device broke through the limitation of previous microchips, which sorted targets based on individual cell or particle properties. Moreover, label-free cells significantly simplified the process of sample preparation. As shown in Figure 1c, a deterministic lateral displacement (DLD) array in a direct current (DC) field first separated the analytes on the volume and surface charge difference. Then, a bipolar electrode (BPE) well aligned the particles into a particle beam in the microchannel for subsequent acoustic separation based on the compressibility and density. In parallel, a dielectrophoresis (DEP) force from BPE was used to separate non-viable and viable cells depending on the dielectric properties, and the viable cells were trapped on the BEP edges with a positive DEP force while the non-viable cells were repelled towards the center of the channels and washed away [50].

Ivo et al. not only applied the BAW acoustophoresis to manipulate particles and cells, but they also used BAW to merge droplets. As shown in Figure 1d, the T-junction and flow-focusing geometries stably generated water-in-oil droplets, and the piezoelectric transducer adhered to the underside of the channel transformed the voltage into mechanical vibration for sample manipulation. In the aspect of droplet fusion, BAW acoustophoresis focus-aligned two adjacent droplets on the centerline of the channel and induced two droplets to merge. BAW acoustophoresis sorted target droplets at an extremely high speed via turning on/off the transducer. They also successfully transferred the droplets from a current continuous phase to the another one using acoustophoresis, which contributed to cell washing and medium change [51].

#### 2.1.3. Electric Fields and Microfluidics

The integration of electric fields and microfluidics has existed for a long time, such as on-line electrophoresis, dielectrophoresis, electroosmosis, and electric impedance analysis [52]. The electric field has been regarded as one of the most popular and efficient nonmechanical pumping and separation mechanisms in microchips [53]. However, the electric current generated from the electric field may result in Joule heating, which might lead to irreversible thermal damage to the cells [54].

An aqueous two-phase system (ATPS) was built and used as a liquid filter by one group. The target cells and polystyrene (PS) particles were selectively separated at the liquid–liquid interface with an external electric field. The factors affecting the particles on the ATPS interface were the strength of the electric pulse, particle size, zeta potential, and hydrophobicity of the particle [55].

A trans-membrane voltage (TMV) generated from a nearby electric field will temporarily change the permeability of a cellular membrane and the extracellular components will flow into the cells through the cell membrane, which is known as electroporation. Therefore, the reversibility of electroporation provides an effective treatment option for tumors. As shown in Figure 1e, the integration of electroporation and microfluidics provided a platform for the rapid and precise identification of bacterial strains. The electric field strength for electroporation of different types of bacterial strains were demonstrated in this paper: *Escherichia coli* BL21 (3.65 ± 0.09 kV/cm), *Corynebacterium glutamicum* (5.20 ± 0.20 kV/cm), and *Mycobacterium smegmatis* (5.56 ± 0.08 kV/cm) [56].

#### 2.1.4. Optical Fields and Microfluidics

Due to the prominent advantages of optics-based detection, including non-invasiveness, easy integration, rapid response, and high sensitivity, many optical detection methods have been incorporated into microfluidic devices, such as fluorescence, absorbance, colorimetry, chemiluminescence, and scattering [57]. Among them, fluorescence detection is still the most widely used because of its high sensitivity, high selectivity for cellular and molecular sensing, and low backgrounds [58].

The integration of microchips and optical systems can also efficiently and precisely sort targets. Fluorescent-dyed particles were excited with an integrated optical waveguide network within microchannels. A diode-bar optical-trapping scheme guided the particles across the waveguide/microchannels and selectively separated the particles based on their fluorescent intensity [59].

In addition to fluorescent detection, absorbance spectroscopy is the most straightforward detection method in separation science. However, this method is significantly constrained by the length of the optical path, which affects the sensitivity of the analysis [60]. The Easley group combined the lock-in technique and absorbance detection to improve the LOD of analytes in droplet microfluidics (Figure 1f). Ultimately, a detection limit of 3.0 × 10^−4^ absorbance units or 500 nM bromophenol blue (29 fmol) was achieved with an optical microscope and a standard, single-depth (27 μm) microchip [61].

**Figure 1 micromachines-14-00972-f001:**
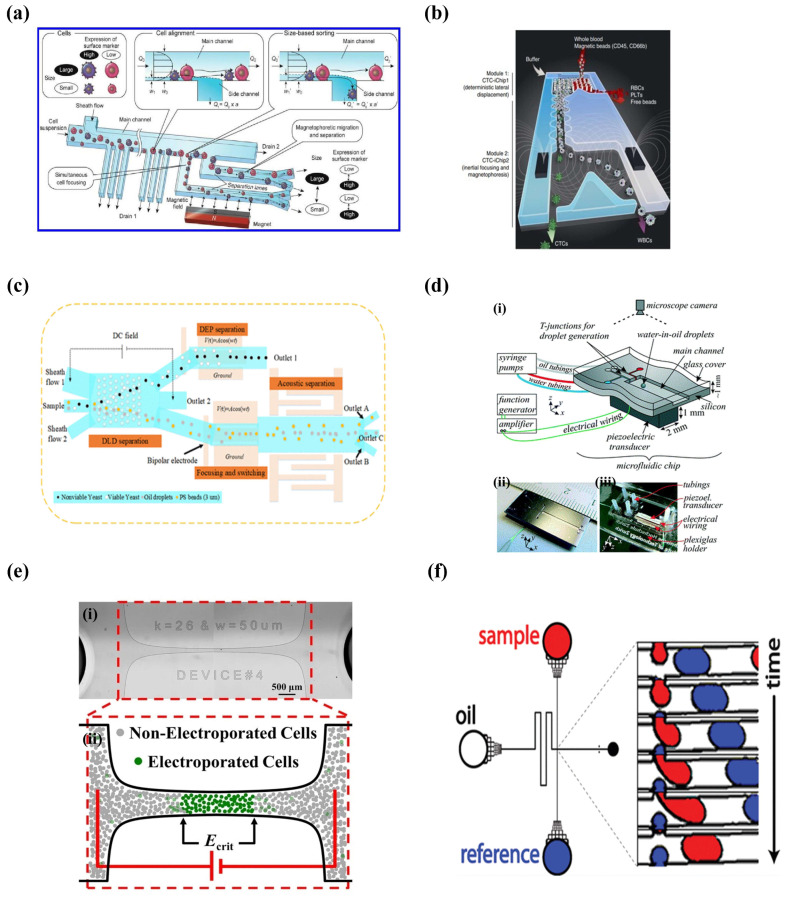
External fields in integrated microfluidic devices. (**a**) Schematic of magnetophoresis-integrated hydrodynamic filtration system. (Reprinted with permission from Ref. [45], copyright 2013 American Chemical Society). (**b**) Schematic of CTC-iChip. (Reprinted with permission from Ref. [46], copyright 2014 Nature Publishing Group). (**c**) Schematic diagram of an integrated microfluidic system designed to achieve multitarget separation. (Reprinted with permission from Ref. [50], copyright 2021 American Chemical Society). (**d**) Sketch of the microfluidic as well as the front and back side of the microchip. (Reprinted with permission from Ref. [51] under a Creative Commons Attribution 3.0 Unported Licence). (**e**) Microfluidic device to determine the critical electric field required for bacterial electroporation. (Reprinted with permission from Ref. [56], copyright 2016 The Author(s)). (**f**) The microchannel layout for alternately generating signal and reference droplets. (Reprinted with permission from Ref. [61], copyright 2012 American Chemical Society).

### 2.2. Introduction of Electrochemical Biosensors

Electrochemical biosensors use electrodes as conversion elements and fixation carriers and fix biological recognition elements (such as proteins, antibodies, enzymes, nucleic acids, cells, etc.) as sensitive elements on the electrodes. The specific recognition between biomolecules generates catalytic or binding activities between the target analyte and the biological recognition element on the electrode, and the final binding reaction is converted into a detectable electrical signal (such as current, potential, resistance, or impedance) through the electrode, which is proportional to the analyte concentration, thus achieving qualitative or quantitative detection of the target analyte [62]. Electrochemical biosensors have the advantages of high sensitivity, portability, low cost, simplicity, and easy operation [63].

Electrochemistry typically uses the three-electrode system: a working electrode (WE) where oxidation or reduction reactions occur and are measured; a reference electrode (RE) that gives a known potential to the redox reactions that happen and precisely controls the potential on the WE; as well as a counter electrode (CE) that forms a series circuit with the WE and acts as a conductor of electricity [64]. Although electrochemical sensors have many benefits, as mentioned, traditional electrochemical detection consumes large volumes of reagents and requires bulky potentiostats, macroelectrodes, and other large equipment [65]. The main components of electrochemical biosensors are electrodes, which are suitable for miniaturization, batch microfabrication, and integration with other components on a single microchip. The applications of electrochemical sensors in biological and chemical detection in microfluidic systems has many benefits, including a low volume of reagents, less detection time, high signal-to-noise ratio, and low consumption of metal materials [66]. To adapt to the microchip size, micrometer-scale “microelectrodes” and “ultramicroelectrodes” have been developed recently and become popular in microfluidics. The synergistic integration of electrochemical sensors and microchips in a signal platform helps researchers discover unknown bioanalysis phenomena. The specific properties of liquid metals (simple fabrication, easy integration, stretchability, reconfigurability, low power consumption, etc.) yield advantages in their integration with microfluidics. For example, some liquid metals can be used as electrodes for microchips because of their high conductivity and fluidity [67]. Kong et al. proposed a novel technique for fabricating three-dimensional (3D) multilayer liquid-metal microcoils together with the microfluidic network through the lamination of dry adhesive sheets, which is beneficial for cleaning up the microfluidic chips and avoiding cross-contamination when reusing the microcoils. A nuclear magnetic resonance (NMR)-based biosensing system integrated with microcoils was successfully applied to aid the diagnosis of anemia [68].

The applications of electrochemical sensors in microchips offer tremendous advantages for facilitating the evolution of modern micro-total analysis systems (µTAS), such as inherent miniaturization, low power requirements, low limit of detection (LOD), and compatibility with advanced micromachining systems [69]. The development of LOC or μTAS mainly relies on semiconductor microfabrication technologies, but the MEMS and microelectronic IC fabrication technologies determine the advancement of micro/nano fabrication technologies. Therefore, the MEMS and IC fabrication have huge effects on the electrochemistry-based LOC and μTAS. There are some challenges, though the concept of sensor–fluid integration has been proposed for long time. The mismatch of footprint between the microfluidic device and the CMOS chip is one of the stumbling blocks, and the other difficult points are the direct contact between the sample fluid and the electrodes and the topographical conflict between the electrical interconnect and the microfluidic channels. Huang et al. proposed an integrated system of CMOS microchip and electrochemical sensors. This platform contained microfluidics, electrode arrays, and CMOS ICs, which introduced both silicon substrate carriers to expand the surface area beyond the CMOS chip and resolved topological conflicts between electrical interconnects and microfluidic channels. This system brings the electrochemical advantages to help the general public without compromising accuracy and reliability [70]. Therefore, electro-microchips have become one of the most popular research directions in μTAS.

### 2.3. The Protocols and Applications of Electrochemical Microfluidics

Microfluidics-based electrochemical biosensors, as a new application area, may be an ideal answer for developing next-generation portable analysis systems [71]. At present, many academic studies have reported various electrochemical analysis protocols in microfluidics. According to the detection principles, five types of common electrochemical detection techniques and the relative applications in microchips are presented here.

#### 2.3.1. Amperometry

Amperometry is a kind of electrochemical analysis where a constant potential is applied to the WE and the RE, and the current generated by the oxidation or reduction of electroactive substances at the working electrode is directly monitored, with a linear relationship between the magnitude of the current and the concentration of the analyte.

Early microfluidics-based amperometry detection systems were mainly used for electrophoretic separation and detection. Woolley et al. developed an integrated platform that contained microfluidic devices, capillary electrophoresis, and amperometry detection for the first time [72]. Currently, microfluidic amperometry sensors (MAS) are widely used in medical diagnostics and health care. The MAS invented by Kaur et al. achieved highly sensitive measurement of cholesterol with an LOD of 0.10 mM over a wide concentration range, and the device had the potential to be added to POC diagnostic devices via inserting the electronic circuits [73]. For the first time, Senel et al. proposed a novel MAS for quantifying dopamine (DA) levels in cerebrospinal fluid (CSF) and plasma in a Parkinson’s disease (PD) mouse model. The microelectrode was surrounded by a ~ 4 × 4 ${mm}^{2}$ micro-chamber and the total volume of the PDMS micro-chambers is 2.4 μL (Figure 2a). The electrochemical oxidation of DA on gold microelectrodes was monitored via the amperometry method at the potential of 0.2 V, and the results demonstrated that this MAS device had the ability to analyze the DA in the range of 0.1–1000 nM. A 50 μm electrode spacing significantly reduced the Ohmic drop and improved the reliability of the analysis [74].

Digital microfluidics (DMF) is a special microfluidic technology with amperometry analysis that enables the manipulation of discrete droplets containing samples and reagents on a flat surface. Specifically, the discrete droplets are manipulated “digitally” on a two-dimensional array of identical unit electrodes. Compared with droplet-based microchips, DMF mixes reagents more uniformly and does not require micro-pumps and microvalves, which greatly simplifies the design and manufacture of the equipment. The absence of microchannels reduces cross-contamination and eliminates dead volume. In addition, because each droplet can be controlled independently, these systems are also dynamically reconfigurable, allowing droplet manipulation to be performed at any location on the array, allowing a variety of processes to be performed simultaneously in a simple and compact design while performing a set of bioassays [75].

DMF couples a complicated sampling process and electrochemical biosensors well. The similar fabrication methods and the ease of integrating electro-sensors and DMF that have an electrode array make the electrochemical sensing mechanism highly compatible with the DMF platform. Wheeler’s team presented the first digital microfluidic electro-immunoassay in a digital microfluidic device by means of electrodeposition without external electrodes, which was used for successfully measuring thyroid stimulating hormones [76]. Later, Wheeler’s team proposed DMF devices coupled with nanostructured microelectrodes (NMEs) for out-of-lab distributed diagnostics with etching sensing electrodes (each consisting of three Au-NEMs, one CE, and one RE), a DMF counter-electrode, five apertures on the sensing electrodes, and a hydrophobic coating in four steps on the top plate of an Indium tin oxide (ITO)-coated slide (Figure 2b). A rubella virus (RV) IgG immunoassay was developed based on this system and analyzed via current change, and the detection limit of this RV assay was 100 times lower than the immunological threshold set by the World Health Organization [77].

#### 2.3.2. Voltammetry

Voltammetry is the most common method of electrochemical analysis, where a potential is supplied between the WE and RE, and the changes in current are also measured. The difference is that the potential changes dynamically over a set range of sweeps in voltammetry and the current response is usually proportional to the concentration of the analyte. There are also various types of voltammetry, such as polarography, differential pulse voltammetry, linear sweep voltammetry, cyclic voltammetry, etc., based on the methods of changing electric potential [71].

TK Dhiman et al. fabricated an electrochemical nanochip based on ${CeO}_{2}-Nps$, which was used for ochratoxin A (OTA) analysis. OTA in the concentration range of 350 ${pg\cdot mL}^{-1}$ to ${ng\cdot mL}^{-1}$ was detected using differential pulse voltammetry (DPV) on a ${CeO}_{2}-Nps$-based immuno-electrode. The immuno-electrode was immobilized with OTA antibodies and BSA, and the OTA solution flowed through the ${BSA/Anti-OTA/CeO}_{2}/OTA$ electrodes with a syringe pump. The specific bond between antibodies and antigens generated changes in the charge on the electrode surface and resulted in a rise of the current peak in the DPV. The micro-size channels of the microchip had superiority in detecting the weak signals of immunoassays compared to conventional electrochemical electrodes [78]. Wheeler et al. proposed the two-plate DMF platform containing droplet manipulation and on-line voltammetric analysis with electrochemical electrodes surrounding four-DMF electrodes (Figure 2c). The LOD of acetaminophen reached 76 μM with a 4% mean RSD using linear scanning voltammetry in this system [79].

In recent years, carbon-based nanomaterials have been used in electrochemical detectors due to their excellent electrical conductivity, structural flexibility, high strength, and other outstanding properties. However, it is still controversial whether there are more suitable materials for fabricating electrochemical sensors compared to graphene and its parent graphite. The effects of several different materials (graphite oxide (GO), chemically reduced graphene oxide (CRGO), graphene oxide (GO’), electrochemically reduced graphene oxide (ERGO), and glassy carbon (GC)) on the electron transfer between hemoglobin in solution and solid electrodes were explored and compared in this work. The electrochemistry of hemoglobin in solution was investigated via cyclic voltammetry and differential pulse voltammetry. The results differed from previous studies in that carbon nanomaterials did not significantly enhance the electron transfer, and the bare glassy carbon remained an appropriate electrode material for electrochemical sensing [80].

Cyclic voltammetry (CV) is the most popular method in voltammetry-based detection. In CV, the potential of a triangular waveform is scanned between two values at a fixed rate and the scan is divided into two parts: a negative scan and a positive scan. During the negative scan, the potential gradually decreases and the electroactive analytes in the electrode are reduced (reduction wave). For the positive scan, the potential gradually increases, and the reduction products are oxidized in the electrode (oxidation wave). The current–voltage curve is called a cyclic voltammogram [81]. Typically, CV is used to measure electrode reaction parameters, determine reaction mechanisms, and observe possible reactions over the entire potential scan range. For a new electrochemical system, CV, also known as “electrochemical spectroscopy,” is generally considered the first priority for evaluation [82]. In addition, CV is used to characterize the performance of different materials or electrodes with different surface modifications [83].

In addition, cyclic voltammetry has been applied for the quantitative analysis of samples. For example, Srikanth et al. developed a microfluidic platform without any biological modification of the electrodes and used CV to study the variation of bacteria concentrations overtime [84]. For integration with DMF, K.C. et al. coupled a three-electrode-based electrochemical system and an electrowetting on dielectric (EWOD) digital microfluidic device to detect iodide droplets via cyclic voltammetry and successfully quantified iodide, which demonstrated that the combination of EWOD microfluidics and electrochemical sensors could achieved a rapid and accurate analysis of targets with a small volume of reagents [85]. Later, Yu et al. proposed an EWOD digital microchip by installing a microfluidic module on the bottom board and a three-microelectrode system on the top board (Figure 2d). A fully automated analysis of ferrocenemethanol (FcM) and dopamine (DA) was finished with cyclic voltammetry with a high sensitivity of 2145 nA$/\mu M{/cm}^{2}$ and a low detection limit of 0.42 μM in the concentration range between 1.0 and 50.0 μM [86].

**Figure 2 micromachines-14-00972-f002:**
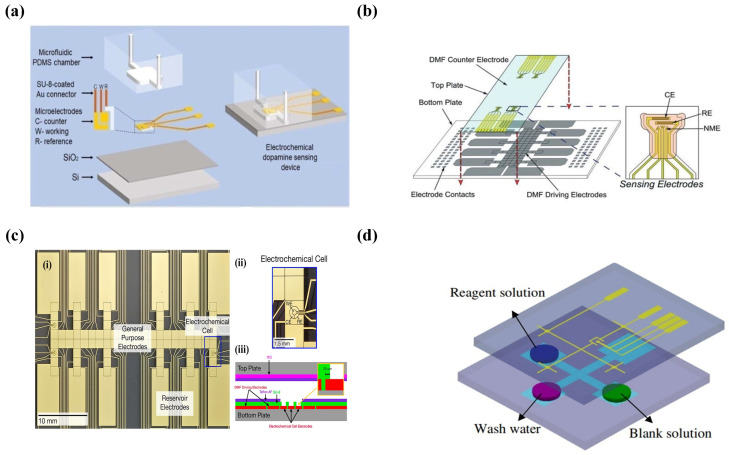
Microfluidics-based amperometry and voltammetry detection systems. (**a**) Schematic of the electrochemical sensing device. (Reprinted with permission from Ref. [74], copyright 2020 American Chemical Society). (**b**) Schematic of DMF devices with integrated nanostructured microelectrodes (NMEs). (Reprinted with permission from Ref. [77], copyright 2015 Royal Society of Chemistry). (**c**) Schematic of double-plate DMF platform and side view of electrochemical electrode. (Reprinted with permission from Ref. [79], copyright 2013 American Chemical Society). (**d**) Schematic of a digital microfluidic biochip for fully automated microarray analysis of ferrocene methanol FcM and DA. (Reprinted with permission from Ref. [86], copyright 2014 IOP Publishing, Ltd.).

#### 2.3.3. Potentiometry

In the potentiometric-detection technique, the charge potential accumulation between the WE and the RE is measured under zero-current conditions. In other words, the analysis of the target analyte is achieved using the potential change between the ion-selective electrode (ISE) and the reference electrode [87]. The potential sensor usually consists of a reference electrode and an indicator electrode. The potential of the indicator electrode varies proportionally to the logarithm of the ion activity, so the potential stability and reliability are key parts of this analytical method.

For any ion-selective electrode (ISE), the most important parameters are the slope, LOD, selectivity, and response time. The slope provides information about the charge interactions between the target compound and the membrane. The sensitivity of the ISE is given by the LOD and the response time is defined as the time it takes for the sensor to reach 95% of the expected response when the analysis occurs [88].

A microchip with an all-solid-state potential biosensor array was developed by Liao et al. Three $P_{t}/C_{r}$ electrodes were fabricated on a glass substrate through the electron-beam evaporation process. The electrodes were further modified with iridium oxide, calcium ion-selective and potassium ion-selective membranes, respectively. The reference electrodes were formed by depositing and patterning Ag/Au/Cr and processing with electron beam evaporation. The detection sensitivity of pH, ${C_{a}}^{2+},$ and $K^{+}are$ 62.62 ± 2.5 mV ${pH}^{-1}$, 53.76 mV ± 3 mV-$log{[K^{+}]}^{-1,},$ and 25.77 mV ± 2 mV-$log{[{C_{a}}^{2+}]}^{-1}$ in this potentiometric method, respectively [89]. Gallardo-Gonzalez et al. report a creative microchip that consisted of a microfluidic device and electrochemical microelectrodes for in situ and real-time measurements of ammonium in flowing water (Figure 3a). In the control experiment, the WE worked as an ammonium-selective electrode, and the performance of this system was evaluated via the potentiometric analysis of the target. The results showed the sensitivity of the system, and the detection limit was 55 mV/decade and 4 × $10^{-5}$ M, and the whole response time was between 10 and 12 S. In addition, the robustness results also showed that this micro-system remained after being immersed in sewage for at least 15 min [90].

Farzbod et al. achieved the installation of potassium-selective sensor arrays on digital microfluidics for the first time (Figure 3b). In this equipment, the $A_{g}Cl$ layer of the RE quickly was dissolved in the sample solution and then the silver-plated solution and hydrochloric acid are sequentially driven onto the sensing electrode on this DMF platform. The potentiometric detection of potassium ions in eight different concentrations of KCl solution is performed in this system, and it takes about 250 s for the EMF (electromotive force) to finish each measurement. The slope of the average EMF data was 58 mV/log, which demonstrates that the electrochemical sensor can be well-integrated with the digital microfluidic platform [91].

Polymer-membrane-based ISE has become the most popular wearable sensor for monitoring the electrolytes in sweat. Many wearable devices do not require a dedicated microchip because sweat can directly flow into the reaction chambers attached on the skin and trigger the biochemical reactions. Sempionatto et al. proposed a potential-sensor-based wearable device that included sensors for detecting $K^{+}$ and ${N_{a}}^{+}$ on the top layer of PDMS, four 2 mm diameter reservoirs, and four channels connected to the detection chamber. This microchip platform with flexible electronics achieved real-time transmission of wireless data to mobile devices and exhibited a selective potential response to $K^{+}$ and ${N_{a}}^{+}$ [92]. Alizadeh et al. reported another wearable device for detecting $K^{+}$ and ${N_{a}}^{+}$, where both the solid ISE and the RE were fabricated on a PET substrate, and then conductive carbon and dielectric insulation layers were printed on the substrate. Finally, a contact transducer layer was electrodeposited on the exposed carbon layer [93].

#### 2.3.4. Conductometry

The conductivity sensor is a miniature two-electrode device for measuring the conductivity of a thin electrolyte layer near the electrode surface. The principle of conductivity-based detection is that the changes in the charge concentrations should lead to a change in the conductivity of the sending layer. Currently, conductivity-based detection is considered one of the simplest electrochemical analytical method in microfluidic devices [94].

Lee et al. proposed a single site-specific polyaniline (PANI) nanowire biosensor to quantify cardiac biomarkers with benefits of high sensitivity, good reproducibility, and high specificity. The electrodes were lithographed and deposited on the silicon substrate with electron beams, and the nanochannels were also lithographed on the PMMA between two electrodes. Finally, the microfluidic channels were adhered to the functionalized PANI nanowires [95]. Venzac et al. reported a microfluidic device analyzing short DNA sequences of *Staphylococcus aureus* via a conductometric method. Figure 3c shows a schematic diagram of the detection system where the conductivity changes were detected from the planar and sample solution. Then, the raw signal was processed using a discrete wavelet to extract the information of conductivity change, and the slow baseline change and electronic noise were eliminated here. Compared to conventional conductivity assays, electrohydrodynamic aggregation-based conductivity assays are more sensitive to small changes in the DNA concentration in the sample solution [96].

Wu et al. reported a digital microfluidic platform for on-chip in situ monitoring of the spore-germination process within *Bacillus atrophyticus* via a conductivity method. The manufacturing process of the chip is as follows: chromium and gold layers were deposited on the glass by sputtering, and then patterned electrodes were etched and patterned. At selected time points, on-chip and off-chip experiments were performed, and no statistically significant differences were found between the germination rates. However, the on-chip conductivity method sacrifices a little detection sensitivity [97].

#### 2.3.5. Electrochemical Impedance Spectroscopy (EIS)

Electrochemical impedance spectroscopy (EIS) is one of the most important electrochemical techniques, and the first publication about EIS dates back to 1975 [98]. Electrochemical impedance spectroscopy originates from the frequency-response analysis in electrical engineering. The basic principle of EIS is using a small-amplitude sinusoidal potential wave as the excitation signal to actively perturb the electrochemical steady-state system and measuring the change in the ratio of the AC potential to the response (current) signal (i.e., impedance of the system). In contrast to other conventional electrochemical methods, EIS is a steady-state technique that uses small-signal analysis to detect signal relaxation over a very wide frequency range, so EIS is able to study intrinsic material properties or specific processes that may affect the conductance, resistance, or capacitance of an electrochemical system [99].

EIS biosensors have many advantages, such as easy operation, fast response, miniaturization capabilities, low cost, sensitivity to analytes, and simplification in integration with microfluidics [100]. Ben-Yoav et al. proposed a microfluidics-based electrochemical sensor that consisted of a patterned electrode chip and a double-layer valve. This EIS-based microchip was used to detect the hybridization between single-strand DNA probes and their complementary ssDNA targets. This device provided programmability and automation for the high-throughput analysis of DNA hybridization and the LOD was 1 nM [101]. Wang et al. proposed a microfluidic device combined with electric impedance flow cytometry (IFC) and EIS for analyzing the electrical properties of single cells (Figure 3d). The IFC and EIS coplanar electrodes were inserted in the main channel and below the capture point, respectively. There are two types of microchannels in this system, a main channel for cells to pass through and side channels for cell trapping. In addition, the flow resistance of the main channel was smaller than that of the side channel. In this platform, the first cell will be trapped in the first capture reservoir, and then the remaining cells must flow in the main channel. In the next cycle, the second cell will be captured, and so on. EIS obtained full impedance spectra containing rich characteristic information of cells, and IFC quickly collected several representative frequency-dependent impedance data points [102]. Three types of cancer cells were analyzed in this platform.

In another study, a system consisting of EIS and DMF for the dynamic analysis of peripheral blood mononuclear cells (PBMC) was introduced. Thin-film transistors (TFTs) and pixel electrodes were integrated in the electrode backplane for independently controlling each pixel electrode in this system. The top ITO plate was modified by etching sixteen 2 mm×2 mm interdigital electrode arrays (Figure 3e). The interdigital electrode arrays combined with EIS enabled the detection of analytes encapsulated in droplets [103]. Liu et al. installed an EIS-based biosensor into a DMF platform for the analysis of peripheral blood mononuclear cell (PBMC) abundance. A gold layer was deposited on an ITO substrate with electron-beam physical vapor deposition, peeled, etched to create IDEs, and finally coated with anti-45 on the gold IDEs to manufacture a biometric layer. The detection of different concentrations of PBMCs in dynamic and static cell culture modes was compared in this system, and the results showed that the dynamic mode had a higher sensitivity than the PBMCs in the static condition [104].

**Figure 3 micromachines-14-00972-f003:**
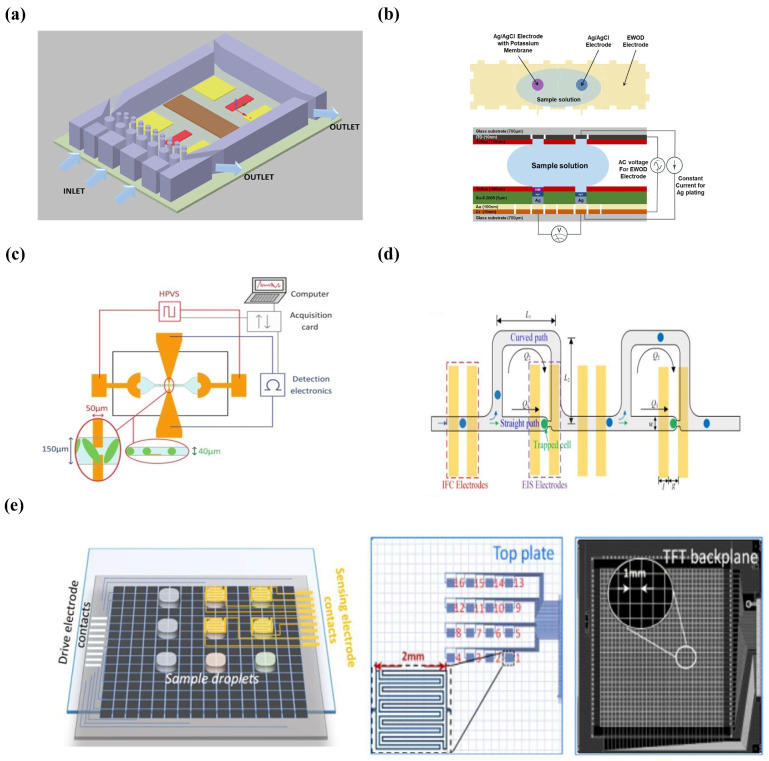
Microfluidics-based potentiometry and conductometry detection systems. (**a**) Microfluidic chip for electrochemical analysis in an aquatic environment. (Reprinted with permission from Ref. [90], copyright 2019 Elsevier). (**b**) On-chip ion-selective electrode calibration procedure. (Reprinted with permission from Ref. [91], copyright 2018 Elsevier). (**c**) Schematic of the system for small DNA sequence detection of *Staphylococcus aureus*. (Reprinted with permission from Ref. [96], copyright 2017 Royal Society of Chemistry). (**d**) Fabrication process of DMF chip for in situ detection of endospore germination. (Reprinted with permission from Ref. [102], copyright 2019 American Chemical Society). (**e**) Schematic of sensing electrodes on AM-DMF chip. (Reprinted with permission from Ref. [103], copyright 2022 IOP Publishing, Ltd.).

### 2.4. Conclusions

A variety of active manipulation methods for samples have been widely integrated with microfluidics until now, such as magnetic, acoustic, electrical, and optical fields. This chapter briefly introduced these techniques and their biomedical applications and presented more information about the combination of electrochemistry and microchips.

Both electrochemical biosensors and microfluidics have their specific benefits, so the rational integration of these two analytical devices will create greater functional platforms. In this synthetic system, microfluidics provides portability, enables additional sample preparation capabilities [105], and reduces the consumption of reagents. Electrochemical sensors are particularly well-suited for combination with microchips because they are the least constrained by the requirements of device miniaturization and scale-up manufacturing compared to other analytical modules [106].

Miniaturizing and integrating several laboratory apparatus in a small facility is viewed as a powerful analytical method with higher efficiency, faster analysis, and lower reagent consumption [107]. Considering the ease of integration of electrochemistry and the miniaturization of microfluidics, the coupling of electrochemical sensors and microfluidics should be an ideal high-throughput and cost-effective platform that can support sample manipulation, detection, and analysis [108]. Therefore, the integration of microfluidics and electrochemical biosensors is conceived as an effective strategy to facilitate µTAS for next-generation LOC platforms.

## 3. Wearable Microfluidics

### 3.1. Introduction

With the development of flexible electronics, materials science, wireless communication, and other related technologies, wearable devices are gradually emerging in the fields of medicine, biology, chemistry, sports, the military, etc. Traditional wearable devices (e.g., smart watches) are used to detect human physical indexes, such as the heart rate, body temperature, and motion tracking [109]. However, some biomarkers in body fluids better display health status than those physical signals, which requires a new generation of wearable devices that have the ability to discover biochemical information at a deeper molecular level [110]. The accumulation of experience of life science and medical research on microfluidics has decisive effects on the emergence of wearable equipment, which mainly integrates various electronic modules, membranes, microchannels, reservoirs, and biochemical sensors.

Microfluidics is an ideal technique that is well-suited for the development of wearable devices. First, microchips are usually fabricated with soft and stretchable materials that can be easily bent and folded without compromising their properties, such as PDMS, acrylics, and hydrogels [111]. Second, microchannels are beneficial to transport and manipulate a small volume of fluid, which may weaken patients’ pain and facilitate the collection of body fluids [112]. In addition, MEMS-based micro-systems can be combined well with flexible electronic modules, patterned electrodes, and sensor modules for biomarker monitoring.

Wearable microfluidics provides efficient platforms for the real-time, continuous, non-invasive monitoring of human bio-signals. The addition of big data and the accumulation of personal medical data into wearable devices will help implement further predictions and complex diagnostics [113].

### 3.2. General Functions

In the light of the requirements, microfluidics can take a more important role in wearable devices, including sample collection, storage, analysis, signal transmission, etc.

#### 3.2.1. Sample Collection

Smooth and efficient sample collection is an important step for the usage of wearable devices and normally consists of three types of methods: invasive, minimally invasive, and non-invasive collection. Sample collection can also be divided into two aspects based on the way the force acts: capillary force and osmotic property.

##### Capillary Force

The capillary effect has been applied to collect and store body fluids in wearable devices without the assistance of peripheral devices. Ma et al. built a wearable device that collected sweat through capillary absorption in biocompatible tubes and the collected sweat was spontaneously transported to hydrophilic microfluidic channels to form a continuous flow [114]. Rogers et al. described a soft microfluidic device adhered to skin that used capillary action and the natural pressure to capture and guide sweat through microfluidic channels and a network of reservoirs [115]. Paper-based analytical devices (PADs) are characterized by their low cost, portability, biocompatibility, miniaturization, point-of-care detection, etc. compared to conventional methods [116]. Wearable devices demonstrate the flexible and real-time biosensor concept, while paper-based diagnostic cards represent the state of the art in terms of integration and functionality. Moreover, PADs provide an alternative platform for spontaneous sample and reagent transport through the capillary force, which avoids purchasing external pumps [117]. Toonder et al. invented a flexible microchip in which the wicking paper absorbed sweat when the device was placed on the skin and then the microchannels and cavities were filled with sweat (Figure 4a) [118]. Yang et al. reported a wearable contact lens that spontaneously delivered tears to microfluidic channels and reservoirs via capillary forces [119].

##### Osmotic Property

Osmotic pressure is also an effective method for sample collection in microfluidic devices. Velev et al. added thin hydrogel discs that were equilibrated in saline or glycerol on the microchip, and these hydrogel discs were attached to a permeable membrane. In this system, the difference in the solute concentration between the hydrogel and sample generated an osmotic drive for fluid entry into the device (Figure 4b) [120]. Later, the same team integrated hydrogel discs and paper-based microfluidics (Figure 4c), and the hydrogel discs directly contacted with the round ends of the paper and the skin. Due to high-pressure penetration, the hydrogel automatically extracted sweat containing biomarkers from the skin surface and then utilized the capillary force to transport the mixed solution from the rectangular paper tape to the evaporation pad for expelling [121].

#### 3.2.2. Sample Transmission

After the sample is successfully gathered, the collected liquid sample flows into the detection area through the microfluidic channels. Sometimes, the sample solution has to be directed to a designated reservoir or the reservoirs need to be filled sequentially. Rogers et al. reported a flexible microfluidics with superabsorbent polymer (SAP) active valves and hydrophobic passive valves. The sweat entered the device from the adhesive layer first, and then the sweat filled the reservoir containing the assays due to the existence of the hydrophobic passive valve in the right channels. Once the reservoir was full of sweat, the sweat would flow to the lower layer, which caused the expansion of the SAP material that closed the corresponding inlets and outlets and forced the excess sweat to flow into the next triple reservoir (Figure 4d) [122]. The same lab designed and installed capillary burst valves (CVB) in their skin microchips to achieve precise and continuous sampling. Those four chronologically marked valves were different in their channel widths and dispersion angles, and their burst pressures (BP) also increased gradually in sequence, which guided the sweat sample flowing into the channels and filled the micro-reservoirs (Figure 4e) [123]. They also applied a similar CVB in another soft microchip where sweat first passed into the inlet area and the micro-reservoirs, and finally the sweat solution was removed through a capillary burst valve. Different valves have different dispersion angles and BPs, which enable precise control of the flowing path of the sample solution [124].

#### 3.2.3. Sample Analysis

Wearable microfluidic platforms select a detection system based on various requirements, and several analytical mechanisms have emerged in recent years, such as optical, electrochemical, and mechanical detection. The operational stability, suitability, sensitivity, selectivity, reliability, and power are key factors to be considered when choosing the optimal method for sample analysis.

##### Colorimetric Detection

The most widely used optics-based detection method in wearable devices is colorimetric detection with some attractive features of cheapness, simplicity, rapidity, and semi-quantitative assessment of biomarkers. It relies on the measurable color changes of the reagents to quantify the concentration of the target analytes [125].

Colorimetric detection enables the simple and rapid quantitative assessment of the instantaneous rate, total loss volume, pH, and multiple biomarkers of sweat. The microfluidic system developed by Rogers’s lab had five separate channels and cobalt chloride was embedded in the circular serpentine channel as a colorimetric indicator. Once exposed to the sweat samples, the color would change from dark blue to light purple, which generated information about the amount of sweating. Other channels with four colorimetric reaction areas in the center had the ability to measure chloride, glucose, lactate, and pH. After the cylindrical chamber was filled with sweat, obvious color changes occurred within 1 min (Figure 5a), and finally the corresponding quantitative analysis was performed using UV-Vis spectroscopy and optical graphics [115]. Later, Rogers’s group made further improvements by forming a color reference through a series of in vitro simulation experiments and printed a layer of color reference on the top of the wearable device. What is more, the colorimetric response of each target chemical in sweat over a typical concentration range provided the spectral information for the color reference markers (Figure 5b) [123].

##### Electrochemical Detection

However, when colorimetric detection is used for point-of-care bio-fluid analysis, they bring out some limitations, such as the inability to obtain a continuous response, the significant influence of sweat turbidity, and the essential requirement of advanced equipment for acquiring high-quality digital pictures for quantitative analysis [126].

Electrochemical biosensors play important roles in wearable systems due to their portability, rapid detection, low power consumption, low cost, and high specificity. These wearable electrochemical sensors promise new opportunities in future medical diagnostics, health care, and more. A variety of electrochemical-sensing principles have been introduced in the previous chapter. In wearable systems, the electrochemical modes of potentiometric and current measurements are mainly used. J.W. et al. showed the first example of continuously monitoring lactate levels in sweat with an epidermal electrochemical biosensor, which demonstrated the real-time kinetics of lactate during exercise. The epidermal sensor consisted of three electrodes and had a high sensitivity with a correlation coefficient of 0.996 in the linear dynamic range. In the practical experiments, the temporal dynamics of lactate during prolonged cycling were recorded using amperometry, and the resulting curve reflected the changes in lactate as the exercise intensity changed [127].

Gao et al. developed a fully integrated wearable sensor array coupled with an amperometry glucose/lactate sensor, and the sensitivities of glucose and lactate were 2.35 nA/μM and 220 nA/mM, respectively. The potentiometric sensors consisted of ion-selective electrodes and PVB-coated reference electrodes used to measure $K^{+}$ and ${N_{a}}^{+}$ with sensitivities of 61.3 mV and 64.2 mV per decade of concentration, respectively [128].

#### 3.2.4. Signal Transformation

The integration of wearable devices and mobile communication technologies for real-time health monitoring, data processing, and storage has become a research/industrial interest. All the collected signals need to be processed, noise-filtered, data-transmitted, calibrated, and read out [110]. Specifically, Bluetooth and NFC as two conventional proximity communication techniques have been widely used in smart devices.

##### Bluetooth

Bluetooth is a wireless technique that enables the exchange of data between devices over a short distance, and it is widely used in consumer electronics. With the help of the MCU’s built-in 10-bit ADC conversion module and its serial communication capability, Gao et al. successfully transmitted data from five sensors (glucose, lactose, temperature, $K^{+}$, and ${N_{a}}^{+}$) to a Bluetooth transceiver. After pairing with Bluetooth, the Bluetooth transceiver transmitted the data to mobile phones for real-time display and storage in the cloud (Figure 5c) [128]. Wang et al. described a wearable sensor for monitoring the dynamics of sodium in sweat. With the help of a miniature wearable wireless transceiver, this device relayed data to a PC via Bluetooth in the form of a serial data stream at 1 s intervals [129].

**Figure 5 micromachines-14-00972-f005:**
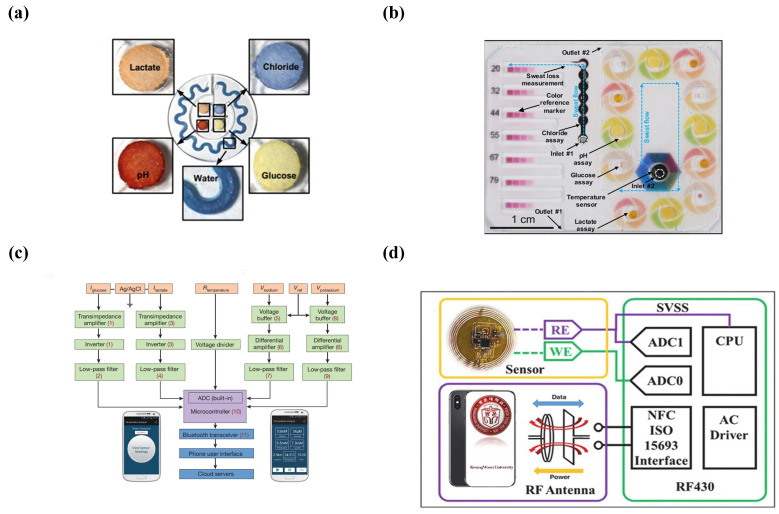
Sample analysis and signal transformation in microfluidics. (**a**) Colorimetric detection of four biomarkers in sweat. (Reprinted with permission from Ref. [115], copyright 2016 American Association for the Advancement of Science). (**b**) Colorimetric sensing microfluidic platform filled with staining solution. (Reprinted with permission from Ref. [123], copyright 2019 American Chemical Society). (**c**) System block diagram of a wearable device for sweat analysis, 1–11 correspond to 11 available integrated circuit components that incorporate the critical signal conditioning, processing and wireless transmission functionalities. (Reprinted with permission from Ref. [128], copyright 2016 Spring Nature). (**d**) System block diagram of a wearable sensing patch for sweat analysis. (Reprinted with permission from Ref. [130], copyright 2021 Elsevier).

##### NFC

In addition to Bluetooth, near-field communication (NFC) is another important tool for signal processing in wearable microfluids. NFC utilizes short-range radio signals to transmit information between two devices, which are typically limited to the centimeter scale. It is not necessary to charge the power-transmission components within the sending equipment, and, inversely, it requires power from the receiving device to complete the measurements and data collection.

Zhang et al. coupled an NFC wireless patch system with flexible electrochemical detectors (Figure 5d). The two-electrode-based electrochemical system acted as a battery to provide power for the NFC chip and a smartphone was used as a radio frequency (RF) power source to wirelessly activate the NFC system. The ADC module in the NFC read and converted the output of the $K^{+}$ sensor, and the programmable gain amplifier (PGA) provided a gain of 2.8 times for the analog output voltage. Finally, the potential value could be used to display the $K^{+}$ concentration after it was read and calibrated on the mobile app [130]. Rogers’s lab installed ultra-thin NFC electronics on top of a microfluidic device that could automatically launch image capture and analysis software when a cell phone was in close proximity [115]. In another type of wearable microfluid, the NFC system was also powered by collecting and transmitting RF. After power-up, the ADC module in the NFC system converted the analog data of the electrodes embedded in the microfluid into digital data, and the data were transmitted back to the phone through the same RF antenna [131].

### 3.3. Applications

#### 3.3.1. Sample Analysis

Although blood is the most popular biological fluid in clinical diagnostics, it is not suitable for continuous monitoring. Currently, researchers are focused on using wearable microfluidics to quantify biomarkers in body fluids. Sweat, tears, and saliva are more attractive samples for biomarker analysis in wearable device because they offer non-invasive sample collection and the ability of in situ monitoring [132].

##### Sweat

Sweat is a fluid containing many biomarkers, such as electrolytes, metabolites, trace elements, and macromolecules, that can provide important information related to human health and physiological status [133]. It is generated inside small glands in human skin and can be collected non-invasively at convenient locations on the body, which makes it an ideal analyte for continuous monitoring. It is wise to place the wearable sensors near the site of sweat production for rapid detection prior to analyte biodegradation [109]. The development of materials science, biochemical sensors, and flexible electronics have laid the foundation for novel wearable platforms for sweat detection at the skin interface, which enables the continuous or intermittent assessment of the biomolecular composition and dynamics of sweat without external devices [134].

Sweat glucose is metabolically related to blood glucose, and real-time monitoring of blood glucose is critical to understanding diabetes progression and disease management [135]. Sweat lactate is a byproduct of the local metabolism of sweat glands that reflects the status of oxidative metabolism and tissue viability. A variety of wearable microfluidics have been developed that can continuously monitor the glucose and lactate levels in real time. Rogers’s team invented a wearable microchip that could simultaneously monitor the sweat rate, pH, lactate, glucose, and chloride (Figure 6a), which consisted of a disposable soft microfluidic network and a reusable thin NFC module. In the glucose sensor, glucose oxidase was dispersed directly in the Nafion and improved the interaction between the glucose and enzymes, which supported the detection of micromolar glucose. In the lactate sensor, the current generated from anodic and cathodic responses were proportional to the lactate concentration [136].

Excessive loss of sodium and potassium in sweat may lead to hyponatremia, hypokalemia, muscle cramps, or dehydration. J San Nah et al. developed a wearable immunosensor with microfluidics and electrochemical sensors for the detection of cortisol biomarkers in sweat based on a ${Ti}_{3}C_{2}T_{x}Mxene-loadedLBG3D$ network of the LOD achieved as low as 3.88 pM with a relative standard deviation of 2.8% for four different sweat samples [137].

Zhang et al. developed a wireless, battery-free, fully integrated wearable system that used a novel ${Ti}_{3}C_{2}T_{x}-MWXNTs$ network to real-time quantify $K^{+}$ in human sweat. The sensitivity of this device was 63 mV/dec, which was amplified to be 173 mV/dec with an NFC amplifier module [130].

Urea and creatinine, related to kidney function, also exist in sweat. Rogers et al. developed a wearable system with capillary burst valves that measured the concentrations of urea and creatinine in sweat (Figure 6b), and the creatinine and urea concentrations in sweat were successfully determined with a simple colorimetric reaction [138].

##### Tears

Tears are a promising fluid for protein, lipid, metabolite, and glucose detection. The biomarkers in tears are diffused directly from the blood, so there is a strong relationship between the biomarker concentrations of tears and blood [139]. However, tear sampling or continuous monitoring is probably uncomfortable and may present a risk of irritation. Capillary micropipettes and swabs are conventional methods for tear sampling. The eye usually reacts in proximity to external objects and some unwanted contact can cause irritation, which cause some difficulty for tear sampling. On the other hand, irritation to the eyes may lead to lower biomarker concentrations in the tear sample [140].

Contact-lens-based wearable devices are an effective solution to address the problem of tear collection. Yang et al. developed a flexible contact lens that spontaneously delivered tears to different microchannels and reservoirs via capillary forces. The inner lens cavity was embedded with a chemical substrate for a colorimetric reaction that could respond to three biomarkers of glucose, chloride, and urea in tears [119]. Kim et al. developed a transparent and stretchable contact lens that allowed sensitive monitoring of the glucose and intraocular pressure in tears (Figure 6c). This method was demonstrated using in vitro detection of glucose in rabbit eyes and intraocular pressure in bovine eyes [141].

Similar wearable device can be combined with glasses in daily life. Wang et al. encapsulated electrochemical detectors in a microfluidic chamber and the micro-chamber was then inserted into the glasses’ nasal pad for monitoring some biomarkers such as alcohol, glucose, and vitamins in tears (Figure 6d). This design avoids the discomfort of placing the device directly in the eye and the potential for infection and vision impairment [142].

##### Saliva

In recent years, saliva has gained popularity as a diagnostic fluid and is a substitute for blood. The saliva is easily gathered and contains biomarkers of several diseases, such as cardiovascular disease, oral and breast cancer, and the human immunodeficiency virus. However, few studies about wearable oral biosensors have been conducted because the abundance of salivary proteins may cause biological contamination [139]. Moreover, the point-of-care saliva sample provides limited physiological insight due to the highly variable composition of last-meal saliva. Despite these challenges, oral biosensing platforms may provide an attractive and painless way to obtain dynamic chemical information from saliva [140]

Lucas et al. made a wearable device for monitoring glucose and nitrite in saliva via 3D-printing technology (Figure 6e). This device prevented any contact between the mouth and reagents and reduced the risk of contamination and the invalidation of reagents. After optimization, the detection limits of glucose and nitrite were 27 ${\mu molL}^{-1}$ and 7 ${\mu molL}^{-1}$, respectively. In the paper, they also found that higher concentrations of glucose and nitrite were detected in saliva from patients with diabetes and periodontitis, as expected [143].

#### 3.3.2. Drug Delivery

Despite the widespread use of drug patches, the efficacy of drug release over time remains a challenge. Wearable microfluidics offers new opportunities for drug delivery. Wearable microfluidics combined with microneedle arrays or microsensors enable accurate, efficient, and safe drug delivery [144].

**Figure 6 micromachines-14-00972-f006:**
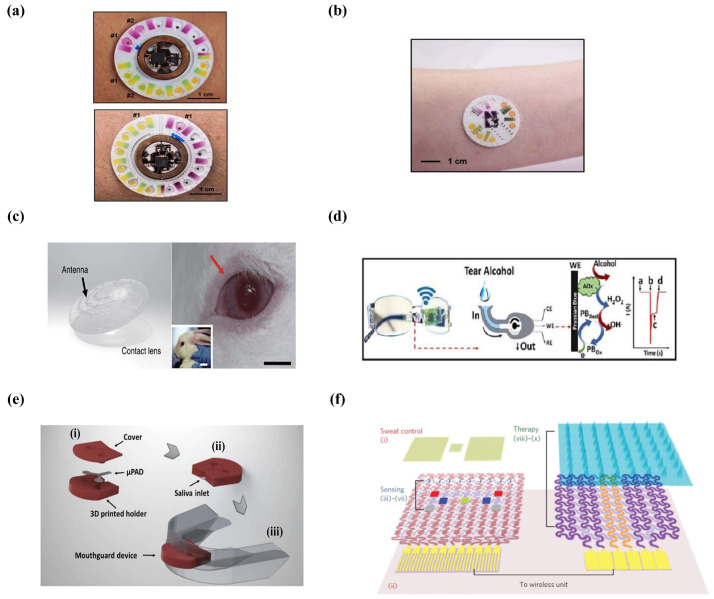
The applications of wearable microfluidics. (**a**) Wearable device for sweat analysis. (Reprinted with permission from Ref. [136] under the Creative Commons Attribution 4.0 International (CC BY-NC 4.0) License). (**b**) Soft microfluidic system for sweat analysis. (Reprinted with permission from Ref. [138], copyright 2019 Royal Society of Chemistry). (**c**) Contact lenses for detecting glucose in tears. (Reprinted with permission from Ref. [141] under the Creative Commons Attribution 4.0 International (CC BY 4.0) License). (**d**) Schematics of the fluidic device and wireless electronics integrated into the eyeglass platform. Where, (**a**) corresponds to the baseline; (**b**) current change due to the captured tear; (**c**) the measured alcohol signal and (**d**) drying of the device. (Reprinted with permission from Ref. [142], copyright 2019 Elsevier). (**e**) Microfluidic devices for saliva diagnostics. (Reprinted with permission from Ref. [143], copyright 2019 Spring Nature). (**f**) Schematic drawings of the diabetes patch, which is composed of the sweat-control, sensing, and therapy components. (Reprinted with permission from Ref. [145], copyright 2016 Spring Nature).

Lee et al. developed a wearable patch for sweat-based diabetes monitoring and feedback therapy. The patch consisted of a sweat-control component, a sensing component, and a therapeutic component (Figure 6f). The microneedle released the drug into the bloodstream, and the microneedle was coated with a thermosensitive PCM that melts at a temperature of 41–42 °C to prevent the microneedle from dissolving when it touched water and effectively prevent the drug from being released prematurely when the temperature does not reach the thermal-response temperature. The multi-channel thermal actuator would control the rate of drug release in a stepwise manner. The system effectively delivered the glucose-lowering drug metformin to mice and rapidly lowered their blood glucose levels [145].

Di et al. made a tensile-strain-triggered wearable device for drug delivery that consisted of a stretchable elastomer and a microgel reservoir. A layer with a microneedle array was attached to the stretchable elastomer. When the elastomer membrane was strained by stretching, the microgel reservoir containing the drug nanoparticles induced compression and released the drug, and the drug further diffused from the microgel reservoir into the microneedles for transdermal delivery. Not only could this method be used for the delivery of anti-infective drugs or painkillers, but they also have successfully used the device to deliver insulin to mice for controlling their blood glucose levels [146].

### 3.4. Conclusions

Since the emergence of mobile devices and smartphones, wearable biosensors have gained tremendous interest and promise to be one of the major advances in wearable health technology. Unlike previously reported wearable devices, wearable microfluidics primarily track physical activity and vital signs and allow real-time rapid detection of accessible biomarkers in the human body, as well as enabling large-scale data collection about an individual’s dynamic health status at the molecular level [139]. However, it is worth mentioning that wearable microfluidics are still in the early stage of development, and most of them are still in the laboratory research stage. Several challenges remain to be solved before large-scale commercialization, such as biocompatibility, biosafety, etc. [147]. On the other hand, the reasonable integration of various components is critical for creating fully functional wearable devices, and the seamless integration of multiple functions into miniaturized components is imperative [112]. It is undeniable that the complementary convergence of microfluidics, flexible materials, biochemistry, and the Internet of Things will promote the continued development of wearable microfluidics.

These developments will transfer biomarker measurements from the central laboratory to the body with the advantages of quickness, reliability, and cheapness [148] Especially in the face of an epidemic, wearable microfluidics are expected to play an important role in monitoring health, physical and mental status, and telemedicine.

## 4. Artificial Intelligence

### 4.1. Introduction

The concept of artificial intelligence (AI) was first introduced in the 1950s [149]. As a branch of computer science, artificial intelligence is the simulation of human intelligence processes with machines, especially computer systems. It enables computers to simulate human behavior and reproduce or even surpass human decisions to solve complex problems independently or with less human intervention [150]. Artificial intelligence (AI) has been a relatively obscure field with limited utility for more than half a century. However, the development of big data and the increased computing power of computers have helped AI become a powerful engine driving the development of all disciplines today [151].

Machine learning (ML), a part of AI, is the core of artificial intelligence [152]. Machine learning (ML) algorithms build a model based on sample data, known as training data, in order to make predictions or decisions without being explicitly programmed to do so. Therefore, the key components of ML are data, algorithms (models), and arithmetic (computing power). Especially in tasks related to high-dimensional data such as classification, regression, and clustering, ML shows good utility and can help produce reliable and repeatable decisions by learning from previous computations and extracting patterns from massive databases [150].

Deep learning (DL) is a relatively new concept that was proposed by Hinton et al. in 2006. A subset of ML, it is viewed as one of the cutting-edge and core technologies of AI [153]. Neural networks are the most important component of DL algorithms. Just as the human brain consists of a network of neurons, a typical neural network also comprises many simple and interconnected neurons that identify hidden correlations and patterns in raw data, then classify and continuously improve them [154]. Unlike basic ML models, algorithms with DL models can automatically extract features through their own neural networks, reducing the requirement of time and labor for constructing feature extractors for each problem [154].

The applications of DL use a layered algorithmic structure called an artificial neural network (ANN), where the leftmost layer is the input layer, the rightmost layer is the output layer, and the middle layer is the hidden layer, whose values are unobservable in the training set. The more hidden layers exist between the input and output layers, the deeper the DL is. Typically, ANNs with two or more hidden layers are called deep neural networks (DNNs), and building complex ‘multilayer DNNs’ allows data to be transferred between different nodes (e.g., neurons) in a highly connected manner [155]. In many applications, DL models perform much better than ML models and traditional data-analysis methods and push AI to a higher level. As usual, DL requires much more data than ML (typically, at least 100,000 samples for common image recognition), and it will show better performance than ML as the amount of data increases.

The important step of AI development is data preparation and collection. Coincidentally, microfluidic technologies can generate highly informative graphics in a high-throughput, cost-effective, and automatic way. Moreover, effective image analysis is regarded as a challenge to most microfluidic experiments. Considering that ML/DL’s powerful analysis of structured data (sequences, images, videos, etc.) can predict complex outputs with unprecedented accuracy, the combination of traditional ML/DL and microfluidics should have the potential to address some previously unsolvable problems.

### 4.2. Machine Learning in Microfluidics

#### 4.2.1. Traditional Machine Learning and Microfluidics

ML methods are classified into supervised learning and unsupervised learning based on whether the input data are labeled or not [156]. In supervised learning, it is essential to understand the relationship between the input and output based on the existing dataset and then train the data to obtain the optimal model. In other words, the supplied training data should have both features and labels in supervised learning, and the machine can find the connection between features and labels by itself through training. Finally, it can determine the labels when facing the data without labels. Conventional supervised learning tasks contain classification and regression, and typical algorithms consist of SVM, KNN, linear regression/logistic regression, decision tree, random forest, etc. [149].

In many practical applications, there is not a large amount of labeled data to be used and it is very difficult to label the data, as it requires a large amount of manual work. Therefore, unsupervised learning will become more important in the long run [157]. As opposed to supervised learning, unsupervised learning looks more like self-learning, and the machine must classify the data without any prior information. The computer will find patterns and associations once it determines they fit well with the raw data, which often produces unexpected results. A typical example in unsupervised learning is clustering. The purpose of clustering is to group similar things together, and the type of the class does not matter much [149]. Thus, an unsupervised learning algorithm does not necessarily have an explicit result.

The reinforcement learning (RL) approach is a trial-and-error method that allows a model to learn using feedback from its own behaviors. It is the closest attempt to model the human learning experience because it can learn not only from data, but also from trial and error [149]. In RL, the computer receives “positive feedback” when it correctly understands or classifies data, and “negative feedback” when it fails. By “rewarding” good behaviors and “punishing” bad behaviors, this learning method reinforces the former one.

##### Supervised Learning

Among the biological applications of intelligent microfluidics, cell classification, including its derivatives (such as cell screening, sorting, counting, etc.), is the most popular research interest and requires processing large amounts of image data. Supervised learning should have the ability to powerfully process the data collected from microfluidics.

Supervised learning of conventional ML improves the accuracy and efficiency of cell classification, as well as reducing the labor. Guo et al. proposed to use an optofluidic time-stretch quantitative-phase microscope and glass microfluidic devices to screen cells. This microchip mainly consisted of four independent microchannels, an orifice layer, and a channel layer. A sequence minimum optimization algorithm was used to train a support vector machine (SVM), which was applied to characterize a heterogeneous population of *Euglena gracilis* under both nitrogen-sufficient and nitrogen-deficient culture conditions. SVM achieved a high-throughput fluorescence-free labeling screening of cells with an error of 2.15% [158].

In the same group, Jiang et al. utilized a similar microfluidic chip-based optofluidic time-stretch microscope combined with ML for detecting platelets in blood. They fabricated a hydrodynamically focused microchip that enabled identifying and sorting the flowing blood cells. The simplest linear classification based on a standard logistic regression model (LR) was used to classify cells from the clear images provided by the microscope, and over 100 features were extracted. An average specificity and sensitivity of 96.6% was achieved in the distinguishment of aggregated platelets, individual platelets, and leukocytes (Figure 7a) [159].

Singh et al. introduced an inline digital holographic microscope coupling with ML for the detection of tumor cells in blood. A collimated laser beam was used to illuminate a sample containing flowing cells in a transparent microchannel, and the forward scattered light from the cells interfered with the non-scattered light to produce a two-dimensional hologram. The hologram was then magnified with the microscope and imaged on a CCD. The micro-size channels (width: 1000 µm, height: 350 µm) maximized the accuracy and precision of imaging. Three features, D (cell size), I_max (intensity of a single brightest pixel (2 × 2 ${\mu m}^{2}$) in a given cell image), and I_mean (average intensity of a region centered on the brightest pixel (6 × 6 ${\mu m}^{2}$)), were extracted from a dataset consisting of erythrocytes, peripheral blood mononuclear cells (PBMC), and tumor cell lines to develop a classifier based on the classification and regression tree (CART) algorithm (Figure 7b). The tumor cells were identified with a false positive rate of 0.001% [160].

Xu et al. collected SERS spectra of two breast cancer cell lines and normal cell lines on a dynamic liquid surface-enhanced Raman scattering (SERS) platform incorporating a microchip, and the SERS spectra were used as a training set for ML. A hose used as a fluidic microchannel was embedded into a 3D-printed recess and a T-connector was used to connect the hose and the pump (Figure 7c). In addition, the classification model for spectrum analysis was built based on the K-nearest neighbors (KNN) algorithm. The sensitivity and specificity of identifying the three types of cells was higher than 83.3% and 91.7%, respectively. In addition, the accuracy of cell identification was 92.8%, 94.4% and 95%, respectively [161].

In addition to cell classification, the combination of supervised learning and microfluidics has been applied to biomarker detection. Manak et al. introduced an ML-assisted micro-device to quantify live-cell phenotypic biomarkers with a single-cell resolution and helped standardize the measurement of biomarkers. The dataset was divided into training and test sets at a ratio of 7:3, and the biomarker rankings were trained with a random forest classifier based on the accuracy of adverse pathology prediction. After establishing the best biomarker ranking, the remaining 30% of the dataset in the sample was classified at the cellular level and sample level to determine its possibility of specific adverse pathology for the risk-stratification prediction of cancer patients [162].

##### Unsupervised Learning

Wang et al. developed an analytical platform with unsupervised learning algorithms for monitoring the biomarkers secreted from single cells. They designed two types of microfluidic chips for single -cell printing and array-containing captured-antibody microprinting, respectively. The secretion data of more than 5000 individual tumor cells were analyzed with the K-means algorithm. Different K values were tested from large to small until all the subgroups were distinguishable from each other, which eventually led to an identification accuracy of 95.0% for tumor cell classification (Figure 7d) [163]. Alvarez et al. proposed a new method for quantifying contaminated and filtered droplets in single-nucleus RNA-sequencing experiments. K-means was used to cluster the droplets to initialize the parameter α and π of EM, and then EM was applied to estimate the parameters of the model to classify the population fragments and cell-based droplets [164].

Another traditional type of unsupervised learning is dimensionality reduction, which looks much like compression and reduces the complexity of the data while preserving as much relevant structure as possible. Desir et al. observed seven different flow patterns (segmental plug flow, droplet flow, segmental plug flow, parallel flow, annular flow, dispersed flow, and irregular flow) in four different biphasic systems in capillary microchannels with laser-induced fluorescence. Using principal component analysis (PCA) to reduce the dimensionality of the potential features of the flow patterns, six important features were eventually identified. A decision-tree model was developed based on the six features, which predicted the flow patterns with an accuracy of 93% (Figure 7e) [165].

Peng et al. demonstrated a POCT system based on NMR. This group used this device to rapidly phenotype label-free molecules with multidimensional inverse Laplace decomposition techniques. The molecular fingerprint of a single drop of blood could be obtained in several minutes on this platform. In this work, multiple unsupervised machine learning algorithms (multidimensional scaling (MDS) and hierarchical clustering) were also introduced for the image analysis of molecular fingerprints, which transformed the complicated NMR correlation maps into user-friendly information [166].

**Figure 7 micromachines-14-00972-f007:**
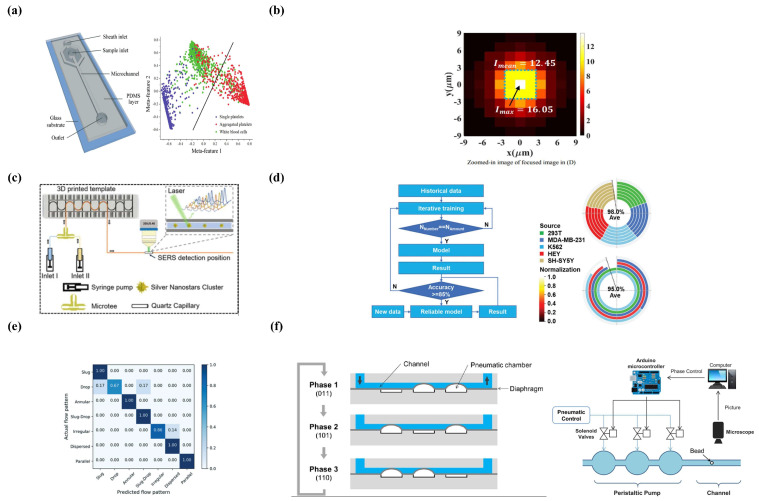
Machine learning in microfluidics. (**a**) Schematic of the microfluidic device and result of cell sorting. (Reprinted with permission from Ref. [159], copyright 2017 Royal Society of Chemistry). (**b**) Heat map of the in--focus image of a single cell. (Reprinted with permission from Ref. [160], copyright 2017 Royal Society of Chemistry). (**c**) Schematic of dynamic liquid SERS platform based on soft tubular microfluidics. (Reprinted with permission from Ref. [161], copyright 2019 American Chemical Society). (**d**) Training flowchart of machine learning and final accuracy. (Reprinted with permission from Ref. [163], copyright 2022 John Wiley and Sons). (**e**) Prediction results for 7 different flow patterns. (Reprinted with permission from Ref. [165], copyright 2019 Royal Society of Chemistry). (**f**) Example of logic state of micro--pump and the structure of the learning system. (Reprinted with permission from Ref. [167], copyright 2021 AIP Publishing).

##### Reinforcement Learning

RL can provide intelligent control for microfluidics. Abe et al. investigated the application of RL algorithms in a micro-peristaltic pump and defined the components of a Markov decision process adapted to a micro-pump. In this algorithm, the flow obtained using micro-valves at the state transition was defined as a reward to obtain a flow-maximizing drive sequence. The micro-pump consisted of three diaphragms and thus, eight logical states (000~111) existed. Figure 7f illustrates the learning system: micro-beads were placed in the observation area and the flow rate was measured using the distance traveled by the micro-beads. The traveled distance was viewed as a reward function and the microcontroller implemented the pumping sequence based on the computer’s calculations. The final three-phase sequence of (000) (110) (001) was obtained, and the flow rate was more than two times higher than that based on conventional sequences [167].

Liang et al. found that RL could assist the digital microfluidic biochips (DMFBs) in providing reliable fluid control while being able to supervise the health of the electrodes and prevent electrode degradation. The system diagram is shown below: a CCD camera was used to capture the droplet positions in real time and a controller connected to the DMFB was utilized to load all bioassays for the droplet-routing task. The experimental results demonstrated that the RL droplet router could learn degradation behavior and transmit droplets using only healthy electrodes, even if the electrodes on the DMFB failed [168].

Microfluidic devices often require significant human intervention to ensure their operational stability in long-time experiments. Dressler et al. used two reinforcement learning algorithms to localize the interface of unmixed phases in laminar flow and control the droplet size in segmented flow in the microchip based on Deep Q-Networks and a model-free scenario controller (MFEC). The experimental results indicated that both of these two algorithms performed better than the manual work at different time scales, which highlights the superiority of the novel control algorithms in high-throughput microfluidics [169]

#### 4.2.2. Deep Learning and Microfluidics

The major stage of traditional ML is processing the natural data in their raw form, but DL is particularly good at domains with large and high--dimensional data. That is the key reason why deep neural networks are widely used in many applications that require processing text, image, video, speech, and audio data [157].

##### Supervised Learning

The convolutional neural network (CNN) is one of the most--frequently used architectures for image classification and its name comes from the convolution in linear operations. The CNN consists of convolutional, nonlinear, pooling, and fully connected layers [170], and the best parameters of each layer will be considered for meaningful output and reducing the complexity of the model. With the development of computer hardware, datasets, and models, as well as the emergence of a series of new ideas and algorithms [171], since the early LeNet-5 architecture, many CNN network architectures have been invented (such as AlexNet, VGG, Inception, ResNet, Xception, etc.) that have greatly enriched the CNN architecture system.

Lung cancer is a kind of cancer with the highest incidence and mortality rate, and early detection is the most effective means for the treatment of lung cancer. Hashemzadeh et al. developed a DL--based microfluidic platform for automatic and high-throughput screening of lung cancer cells. The microfluidic device supported cells a on three--dimensional growth platform to keep the cell population similar to the in vivo environment. The sorting accuracy for five different lung cancer cells with the residual learning convolutional neural network ResNet18 was 98.37% and the relative F1 score (a measure of the classification problem) was 97.29%. In the screening of lung cancer cells and normal cells, the accuracy reached 99.77% and the F1 score was 99.87% [172].

J.S. et al. proposed a microfluidic platform that helped simulate the flow conditions of the spleen in the interendothelial gap (Figure 8a). DL algorithms worked to distinguish specific types of rare hereditary hemolytic anemia by studying the deformation characteristics of red blood cells. Microfluidic technology was used to discover the microvascular characteristics at the cellular level, providing an opportunity to study the biomechanical properties of red blood cells. Each region of interest (ROI) was extracted from the video and was converted into a feature vector by the AlexNET DL network based on CNN architecture. In this DL, the irrelevant image features were eliminated through an unsupervised and automatic feature-selection procedure and a support vector machine (SVM) classifier with a linear kernel was applied to construct the classification model. The average efficiency of this platform was 91% and the accuracy of screening RHHA subtypes was 82% [173].

The performance of three different DL architectures—baseline CNN (two convolutional layers), SimpleNet (13 convolutional layers), and CapsNet—was compared in classifying microfluidic trap images (Figure 8b). The training dataset was augmented through a series of affine transformations, which expanded the training set and improved the performance of the convolutional and capsular networks. These three architectures had different advantages and disadvantages in the recognition of different biological classes. The model integrating these three models exhibited higher overall accuracy than the individual models or models combining two or more by combining the advantages of each model and assigning different prediction weights to those models [174].

Lee et al. presented an image-activation-based sorting technique based on a fast DL model under the Tensor RT framework that determined the sorting or not within 3 ms. The microfluidic device was divided into three parts, a flow-focusing zone, a detection zone, and a sorting zone, and there were an actuation channel, a waste channel, and a collection channel in the sorting zone. Since the waste channel was wider than the collection channel, the focused beads/cells flowed into this channel when there was no actuation signal. The beads/cells were guided into the collection channel when there was an actuation signal. The multifunctional I/O device generated a voltage pulse 2.5 ms before the target beads or cells reached the sorting line, and the piezoelectric pulse was amplified to cause the piezoelectric actuator to push water from the syringe into the actuation channel, which sorted the target beads or cells into the collection channel. ResNet18 was selected as the DL model for classification, and the sorting rates were 98.0%, 95.1%, and 94.2% for 15 μm and 10 μm beads, HL-60 and Jurkat cells, and HL-60 and K562 cells, respectively [175].

RNNs are another popular type of neural network with “memory” that can remember previous information and apply it to the current output computation for processing sequential or time-series data. Honrado et al. designed a new microchip with two detection zones that provided two electric fields: one had a uniform height along the channel and non-uniform width along the channel, and the other one had a uniform width along the channel and non-uniform height along the channel. The combination of these two types of electric fields helped eliminate the effect of position ambiguity. Since the signal of the impedance data stream was a time-series, a modified RNN network (LSTM) was used to process the data, predicting the cell size, velocity, and cross-sectional position at 2500 cells/second [176].

**Figure 8 micromachines-14-00972-f008:**
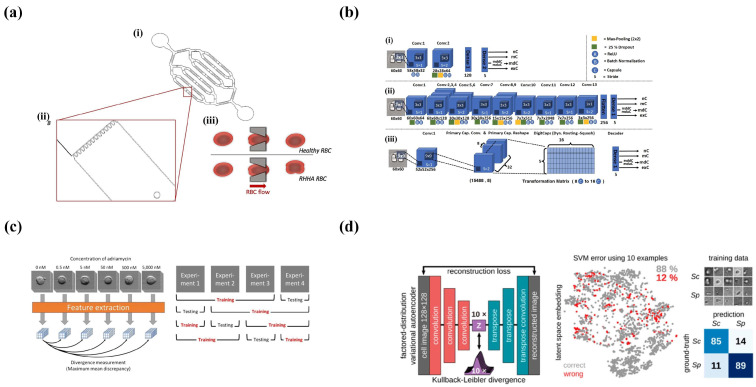
Deep learning in microfluidics. (**a**) Microfluidic chip-design diagram and performance of healthy/RHHA RBC through the slits. (Reprinted with permission from Ref. [173] under the Creative Commons Attribution 4.0 International (CC BY 4.0) License). (**b**) Architectures of three models. (Reprinted with permission from Ref. [174] under the Creative Commons Attribution 4.0 International (CC BY 4.0) License). (**c**) Schematic of the feature analysis and round-robin training and testing. (Reprinted with permission from Ref. [177], copyright 2019 Royal Society of Chemistry). (**d**) VAE architecture for unsupervised learning and result of few--short classification accuracy. (Reprinted with permission from Ref. [178] under the Creative Commons Attribution 4.0 International (CC BY 4.0) License).

##### Unsupervised Learning

Autoencoder (AE) is a popular unsupervised DL algorithm. Kobayashi et al. conducted image recognition of the drug sensitivity of leukocytes via extreme-flux flow cytometry coupled with deep convolutional self-encoders. The small height and narrow width of the channels in the microchip ensured a precise focus and high flow rate for imaging. The combination of a narrow imaging region and wide non-imaging parts avoids the risk of high pressure caused by high-speed flow. The self-encoder extracted features from the cell images and transformed them into a 4,608-dimensional latent space (Figure 8c). The maximum mean difference (MMD) and HSIC could capture dose-dependent morphological changes occurring in drug-sensitive cells, whether cultured cells or primary blood cells [177].

Constantinou et al. developed a self-learning microfluidic platform for single-cell imaging and classification based on a Y-shaped channel and a variational autoencoder (VAE). The encoder was composed of six 3 × 3 ReLU activation convolution kernels, and the decoder was composed of transposed convolutions in the reverse order of the encoder. The encoder mapped the image into a ten-dimensional point in the latent space; each dimension corresponded to an individual feature, and the decoder generated images from these points. Eventually, an SVM classifier was applied in the latent space for few-shot classification with an accuracy of 88% (Figure 8d) [178].

Other unsupervised DL algorithms, such as deep belief networks (DBNs), are also used in microfluidics. DBNs stack multiple independent unsupervised networks (e.g., AE, RBM) and use the hidden layer of each network as the input to the next layer. The DBN algorithm designed by Gopakumar et al. could extract features and sort the localized cell lines without clear segmentation and obvious features. The specific steps are as follows: First, a rectangular bounding box containing the cells are found for rough segmentation, and then the background is subtracted for simple preprocessing. Finally, the bounding box is identified to locate non-repeating cells to format a dataset. Constructing DBN with a restricted Boltzmann machine (RBM) improved the classification accuracy and response rate [179].

### 4.3. Conclusions

ML and DL will have a profound impact on our lives, and all of industry will be changed by the effects of AI. The main focus of Industry 4.0 is technology-driven automation and intelligence in various fields such as smart healthcare, smart cities, and smart business [180]. The combination of microfluidics with ML and DL is an especially innovative approach that should have huge effects on smart healthcare.

The main advantage of droplet-based microfluidics is its high multiplexing capability, which means that it can enable thousands of reactions to react simultaneously and independently, which also introduces the requirement for big-data processing. In recent years, the potential of ML and DL has been explored in microfluidics for biomedical and biotechnological applications. ML has proven to be a useful tool in feature extraction, classification, prediction, and optimization for the large amount of data generated by microfluidic systems [181]. Essentially, the large amount of data collected from highly parallelized microfluidic systems represents an ideal biotechnological application for current DL algorithms [38] The combination of microchips and AI will be automatic and micro-total analysis systems that can effectively solve some tough issues and are capable of predicting results with extremely high accuracy [39]

## 5. Conclusions

In the past decades, microfluidics has shown great promise in enhancing biotechnology applications with small sample volumes, short reaction times, high sensitivity, and high throughput [182]. Microfluidics has the potential of becoming a practical technology that will help daily human life, although there are some challenges. There are four main components in microfluidics, including microchip design and fabrication, microelectronics, sample detection, and data analysis [1]. The success of microelectronics and computer miniaturization has inspired the miniaturization, integration, and intelligence of microfluidics. We are about to enter a special age when everything is connecting by integrating energy, electronics, communication, computers, and sensors. The emergence and improvement of intelligent microfluidics requires advanced materials, electrochemistry, biochemistry, microelectronics, AI, and some other technologies. Intelligent microfluidic systems will provide a powerful platform for biomedical analysis.

Wearable microfluidics is a popular field that contains several types of techniques: microfluidics, biosensors, soft materials, microelectronics, and AI. Wearable microfluidics will be an important tool in medical diagnosis, despite some challenges in data collection, processing, communication, security, and biocompatibility [183]. Wearable electronic devices and digital health based on big-data analytics and ML have great potential to provide real-time diagnostic information to patients.

A combined platform consisting of microfluidics and advanced machine learning tools is inevitable. It will take full advantage of the high throughput and small-volume of microfluidics, as well as the automation and powerful data-processing capabilities of ML [181]. ML not only innovates microfluidic equipment, but also addresses some tough issues in traditional biology and life sciences. ML technology has been shown in facilitating the implementation of AI in microfluidic devices for cell sorting, manipulation, biomolecular analysis, DNA/RNA sequencing, and other biomedical applications. Therefore, we believe that intelligent microfluidics will play a more and more important role in research and industrial fields in the future.

## Figures and Tables

**Figure 4 micromachines-14-00972-f004:**
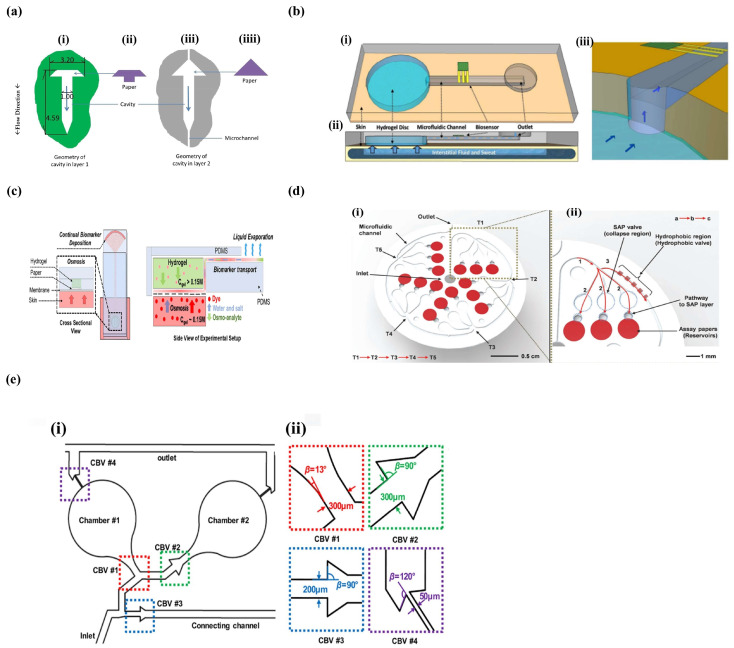
Sample collection methods in microchips. (**a**) Designs of cavities in microfluidic device. (Reprinted with permission from Ref. [118], copyright 2016 Elsevier). (**b**) Schematic of the sweat-collection device. (Reprinted with permission from Ref. [120], copyright 2017 Royal Society of Chemistry). (**c**) Sweat-collection schematic of wearable sweat-sensing platform. (Reprinted with permission from Ref. [121], copyright 2021 American Chemical Society). (**d**) Design of microfluidic channels, reservoirs, and SAP valves for epidermal microfluidic devices. (Reprinted with permission from Ref. [122], copyright 2018 John Wiley and Sons). (**e**) Design schematic of capillary bursting valves. (Reprinted with permission from Ref. [123], copyright 2019 American Chemical Society).

## Data Availability

Data sharing not applicable: no new data were created or analyzed in this study.

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
