# Peer review of "Advances in Integration, Wearable Applications, and Artificial Intelligence of Biomedical Microfluidics Systems"

_micromachines, 2023, doi:10.3390/mi14050972_

Round 1
Reviewer 1 Report
Comments to the Author:
In this manuscript, the authors reviewed some classical electrochemical biosensors and wearable devices using microfluidics. They also further introduced smart microfluidics with artificial intelligence technologies. This work is useful for the community. However, i have several concerns before this manuscript can be accepted. Therefore, in its current form, revisions are needed.
1. The advances in integration for biomedical microfluidics is not well elaborated, since they just discussed the integration of electrochemical sensors with microfluidics. However, an integrated microfluidic device incorporates many of the necessary components and functionality of a typical room-sized laboratory, such as multiple fluidic, electronic and mechanical components or chemical processes, onto a single chip sized substrate. The authors should make a more detailed discussion about this aspect. For example, 10.1021/ac303336f, 10.1038/nprot.2014.044, 10.1021/acs.analchem.1c00312
2. In the wearable microfluidics part, the authors should add some discussion about the paper based microfluidics, since paper-based analytical devices (PADs) have recently gained significant attention because they are simple and inexpensive, require minimal sample, and are readily disposable. Such as 10.1039/c8lc00025e
Reviewer 2 Report
This is well-written article on microfluidic. However, there are some missing references, typo and several
statements which are unclear.
1. Suggest some missing citations.
Line 175
https://www.nature.com/articles/srep06209
DOI: 10.1039/c3ra45417g
Line 864
https://www.nature.com/articles/s42003-020-01262-z
DOI: 10.1002/eng2.12383
https://doi.org/10.1038/s41538-022-00173-z
2. Some of the minor typo as stated in the manuscript.
3. Comment about electrode which are made of liquid electrode such as ... which is vital for integration
https://doi.org/10.1039/C7LC00046D
Lab on a Chip 12 (2), 287-294
Reviewer 3 Report
This paper is a detailed summary of biomedical microfluidic devices. Medical applications of microfluidic devices will be of interest to many readers.
One concern is that many research papers are enumerative. Specifically, there are many figures but I don't understand what they represent. The text is also too small to read. I recommend you focus on important studies and put some strength into explaining them.
Reviewer 4 Report
The paper by Ma et al „Advances in Integration, Miniaturization, and Artificial Intelligence of Biomedical Microfluidics System” has ambition of reviewing a very wide area of biomedical research: microfluidics in biomedical research. However, the review concentrates only on three narrow areas of the microfluidics: electrochemical detectors: wearable microfluidics and artificial intelligence.
When the topics are relevant to the miniaturization of the microfluidics the more crucial and determining factor for miniaturization the “world-to-chip” interfacing has completely been left out of their attention. It includes miniaturization of the fluid actuation devices (like miniature pumps or high voltage power supplies), or power supplies for the whole instrument. Authors are aware of such devices (see text in lines 36-37) but nothing more has been said about that.
Electrochemical detectors have received the author’s attention, but nothing has been said about optical detectors (absorbance or fluorescence). Only (naked eye?) colorimetry has been mentioned in connection with the wearable microfluidics. It would be interesting to know if there have been any developments in the miniaturization of optical detectors in recent years.
In conclusion, the authors should revise their text considering referees comments or address explicitly why “world-to-chip” interfacing and optical detector have not been covered in their review.
Reviewer 5 Report
Ma et al. aimed to review recent advances in integration, the miniaturized and artificial intelligence of biomedical microfluidic systems. It is very ambitious. However, the overall quality of the review is not acceptable. The content of the review is not organized logically. It feels like randomly putting unrelated words together without a focus. For instance, the topic of the introduction part is jumping between the concepts of microfluidic, droplet microfluidic, CMOS, point-of-care application, etc. It is tough to find the theme of the review. Next, electrochemical biosensors were introduced. But, it is not organized with a main focus, for example, the novelty of the microfluidic design, the sensing materials, or the working principle of those reviewed works.
Overall, this is a poorly organized review. The presentation style undermines the significance of this work.
Round 2
Reviewer 1 Report
The authors have addressed all my concerns, so I would like to recommend the paper for publication.
Reviewer 4 Report
The authors have made many changes in their manuscript which have improved its quality significantly. Reviewers recommendations have taken into account and text have been changed where necessary,
Reviewer 5 Report
I am sorry to say that the revision is not convincing. The content of choice is too broad and lacks logical organization.